# Oral Hygiene Practices and Knowledge among Adolescents Aged between 15 and 17 Years Old during Fixed Orthodontic Treatment: Multicentre Study Conducted in France

**DOI:** 10.3390/ijerph19042316

**Published:** 2022-02-17

**Authors:** Camille Inquimbert, Celine Clement, Antoine Couatarmanach, Paul Tramini, Denis Bourgeois, Florence Carrouel

**Affiliations:** 1Department of Public Health, Faculty of Dental Medicine, University of Montpellier, 34000 Montpellier, France; paul.tramini@umontpellier.fr; 2Laboratory “Health Systemic Process”, UR4129, University Claude Bernard Lyon 1, University of Lyon, 69008 Lyon, France; celine.clement@univ-lorraine.fr (C.C.); denis.bourgeois@univ-lyon1.fr (D.B.); florence.carrouel@univ-lyon1.fr (F.C.); 3“Interpsy” Laboratory, University of Lorraine, EA 4432, 54015 Nancy, France; 4Department of Public Health, Faculty of Dental Medicine, University of Nancy, 54000 Nancy, France; 5Faculty of Dentistry, University of Rennes, UMR 6051, CHU Rennes, 35000 Rennes, France; antoine.couatarmanach@univ-rennes1.fr

**Keywords:** oral health, adolescent, knowledge, practice, behaviour

## Abstract

The aims of this study were to assess oral health knowledge, attitudes, and practices among orthodontic patients between the ages of 15 and 17 years old compared to adolescents without orthodontic treatment. This cross-sectional study included 392 adolescents drawn from various French teaching hospitals. A closed-ended questionnaire was used to collect data. Adolescents undergoing orthodontic treatment had a higher knowledge of oral health than adolescents without orthodontic treatment. The majority of adolescents for both groups (69%) claimed to brush their teeth twice a day. Regarding complimentary dental material, 81.9% of adolescents without orthodontic treatment never used an interdental brush and 78.8% never used dental floss. For those undergoing orthodontic treatment, 48.5% never used an interdental brush. Only 4% of adolescents without and 3% of adolescents with orthodontic treatment never consumed fizzy drinks, 4.9% and 3% never consumed sweets, and 4% and 8.4% never ate fast-food. Adolescents without treatment consumed more sodas (*p* = 0.048) and more fast food (*p* = 0.029). Adolescents had insufficient knowledge of oral health. Health education programmes should be implemented to improve adolescents’ knowledge and individual oral prophylaxis with interdental brushes.

## 1. Introduction

Orthodontic treatment is often performed on teenagers, and its prevalence may range from 10% to 35% in developed countries [1]. It usually involves fixed or removable appliances for the correction of mild to severe malocclusions. A systematic review found that orthodontic treatment usually lasts an average of 20 months, with a mean number of required visits at 17.8 [2]. During this period, orthodontic appliances may increase biofilm accumulation and plaque retention, and inadequate oral hygiene can lead to permanent damage to dental tissue, caries, or periodontal lesions. Moreover, orthodontic patients demonstrate higher proportions of Gram-negative species [3], resulting in more inflammation and bleeding [4]. Previous studies [5,6] have shown that orthodontic care is often associated with oral diseases due to excessive plaque retention adjacent to brackets and attachments. Therefore, adherence to prophylactic practices is essential to maintaining good oral health during orthodontic treatment.

Adolescence may increase the risk of dental diseases, as it is a period during which oral care habits are being established, with lower motivation regarding good oral hygiene maintenance [7,8,9,10]. Moreover, adolescents tend to consume more snacks and beverages between meals. For all these reasons, oral care plays a key role [11,12,13] in adolescents becoming at increasing risk for caries [14,15,16] and early periodontal disease [17,18,19].

Currently, we know that maintaining oral hygiene in interproximal spaces calls for special devices, the use of interdental brushes being the most effective method for interproximal plaque removal [15,16]. The daily use of interdental brushes can reduce inflammation and lead to the reestablishment of symbiotic interdental microbiota [17,18].

Changing behaviour is a very long and complex process that takes place over time and at different stages of life. Health literacy has been found to be a strong predictor of individual health, health behaviour and health outcomes [19,20]. Health literacy is a known mediator between socio-economic factors, health behaviour and oral health outcomes in various populations, explaining gradients in oral health status and outcomes [21]. An increase in oral health knowledge is often associated with an increased awareness of oral health and better oral hygiene practices [22].

Although many studies on oral health knowledge and the practice of preventive measures have been conducted among children, few studies are available among adolescents [23,24,25,26,27,28], and epidemiological studies on oral health status and dental knowledge are scarce among French adolescents [29,30]. 

Since adolescents undergoing orthodontic treatment receive more specific information and adequate hygiene recommendations at each regular orthodontic visit than adolescents without orthodontic treatment, the hypothesis of this study was that orthodontically treated adolescents would have better dental knowledge and even more adequate oral hygiene practices than non-treated adolescents. Therefore, the objective of this study was to evaluate the oral health knowledge, attitudes, and practices amongst a sample of 15–17-year-old French adolescents with and without orthodontic treatment.

## 2. Materials and Methods

### 2.1. Study Site and Population

A cross-sectional questionnaire-based survey was conducted from September 2018 to December 2018. A convenience sample was recruited from four teaching dental hospitals (public health and orthodontic departments) located from the north, east, west, and south of France. To be included in this study, (i) the patients had to be at least 15 years old and at most 17 on the day of the inclusion, (ii) they had to have a good understanding of the French language, (iii) the study’s terms had to have been accepted by one of their parents and by the patients, (iv) and the written informed consent had to have been signed by one of their parents and also by the patients. 

The exclusion criteria were: (i) patients who were mentally or physically challenged, and (ii) patients in need of urgent care, because they were not fit for answering the questions in the survey.

Two groups were set up, one for adolescents undergoing orthodontic treatment (OT) and another for adolescents without orthodontic treatment (NT).

### 2.2. Ethics

The protocol and study design were approved by ethics and regulatory agencies and were implemented in accordance with provisions of the Declaration of Helsinki. The appropriate Committee (Local Research Ethics Committee, Montpellier, France) approved the protocol on 10 January 2018 (2017_CLER_12-01). Informed consent had to be signed by all adolescents and one of the parents or legal representatives. The informed consent form contained the name and affiliation of the investigator, a plain language description of the study (intervention), the approximate duration of the interview and the ethics committee approval.

### 2.3. Measurement Tool

A questionnaire was established, comprising 36 questions (see Appendix A). It was drawn from the ESCARCEL survey, where a validated questionnaire was previously developed in different languages [31]. All questions were closed-ended questions. The patient’s knowledge, attitude, and practices (KAP) were assessed by using a questionnaire which included four parts. The first part referred to general information: year of birth, gender, the father’s and mother’s occupations and social security coverage. The second part concerned the characteristics of the dental consultation. The third part described oral health, oral hygiene practices, dietary habits, and tobacco, drugs, and alcohol consumption. The last section of the questionnaire dealt with knowledge, in particular questions about oral health knowledge on hygiene, eating habits and the role of fluoride. Responses on self-perception of general and oral health and on dietary and hygiene habits were made on a four-point scale: “Often” (more than 3 times a week), “Sometimes” (one or two times a week), “Rarely” (less than once a week), and “never”. The response “I don’t know” was selected in cases where the adolescent did not know the answer or did not wish to answer the question.

Adolescents were interviewed face-to-face by the dental practitioners who took part in the survey. To avoid potential information bias, interviewers explained the purposes and confidentiality of the survey and explained that the study had no impact on participants’ examinations. The data was collected on a sheet of paper (printed questionnaire) with an anonymity number (allocation of a random number).

### 2.4. Statistical Method

Sample size: With a confidence level of 95%, a confidence interval of 5% and an average of 50% for any unknown percentage of the questionnaire, and considering the total number of adolescents in France (approximately 2,000,000 according to INSEE data in 2017), it was necessary to include at least 380 subjects in this study, with approximately 50% in each group (orthodontic treatment/no orthodontic treatment). 

Statistical tests: Categorical variables were described by frequencies or percentages and quantitative variables using means and standard deviations. The relationships between oral health knowledge and other qualitative variables were analysed using chi-square tests. Student tests or Wilcoxon tests were used for quantitative variables, depending on the normality of these variables. Multivariate analyses, such as factor analysis, were used to analyse the relationships between different items of the questionnaire. Statistical significance was assumed when *p* < 0.05. All analyses were performed with the statistical software Stata 16.1 (StataCorp, Texas, USA).

## 3. Results

### 3.1. Patient Characteristics

The total sample comprised 392 adolescents, with 226 (57.7%) in the NT group versus 166 (42.3%) in the OT group. The gender distribution was 44.6% (175) males and 55.4% (217) females. The mean age of the participants was 16.43 (±0.51), without gender difference: 16.41 and 16.45 for males and females, respectively (*p* = 0.727). Sociodemographic characteristics of the sample are displayed in Table 1. The fathers’ occupations were mostly administration, or employees, and a quarter were executives. The mothers’ occupations were not statistically different between the two groups (*p* = 0.060) with mostly employees and executives among the OT group. The percentage of adolescents benefiting from medical assistance was significantly higher in the OT group (81.3%) than in the NT group (70.3%), (*p* = 0.013).

### 3.2. Assessment of Adolescents’ Knowledge regarding Prevention of Oral Diseases

The assessment of adolescents’ knowledge regarding prevention of oral diseases are presented in Figure 1 and Appendix A. The proportion of adolescents who knew that fluorides prevent decay was higher in the OT group (65.1%) than in the NT group (46.0%), *p* = 0.001. There was no other significant difference in knowledge between the two groups. Regarding the duration of toothbrushing, 36.1% of adolescents from the OT group thought that the correct answer was between 2 and 3 min, while this percentage was 28.3% in the NT group (non-significant difference, *p* = 0.231).

Almost half of the sample in both groups thought that the role of fluorides was to prevent calculus (48.2% and 37.6%). 

### 3.3. Assessment of Adolescents’ Attitudes and Practices towards Oral Health

#### 3.3.1. Oral Hygiene Habits

The assessment of adolescents’ oral hygiene habits is presented in Figure 2 and Appendix A. The majority of participants from both groups (NT: 88.0%, OT: 91.0%) claimed to brush their teeth at least twice a day. Regarding oral hygiene practice, there was no significant difference between the two groups, except for the use of interdental brushes: adolescents from the OT group used them more often (*p* < 0.001). Concerning at what time the participants brushed their teeth, there were also some differences between the two groups: in the OT group, adolescents were more likely to brush their teeth after breakfast (*p* = 0.019), and after lunch (*p* = 0.020), while in the NT group, they were more likely to brush before breakfast (*p* = 0.018). For brushing after dinner, there was no difference between the groups (*p* = 0.885).

#### 3.3.2. Nutrition and Harmful Habits

The assessment of adolescents’ nutrition and harmful habits is presented in Figure 3 and Appendix A. Fizzy drinks were more often consumed by adolescents without orthodontic treatment (*p* = 0.040). In the same way, fast foods and cigarettes showed significantly higher percentages in the NT group. Sweets were more often consumed by adolescents without orthodontic treatment, but the difference was not significant (*p* = 0.167). High frequency of snacking (5 times a day or more) was significantly increased in the NT group (63.3% vs. 52.4%, *p* = 0.031). Participants claimed that they rarely consumed alcohol and drugs (77.0%), and 96.4% never used alcohol and drugs.

### 3.4. Self-Perception of Oral and General Health

The self-perception of oral and general health is presented in Figure 4 and Appendix A. Several situations and behaviours were significantly different between the two groups. Adolescents from the OT group were more likely to have difficulty chewing (*p* = 0.002). When grouping ‘rarely’ and ‘sometimes’ together, adolescents from the OT group still had more difficulty eating (*p* = 0.030) and suffered from toothache (*p* = 0.041). The presence of orthodontic appliances was related to feeling embarrassed about his/her teeth (*p* = 0.007). The perception of general, oral, or gingival health was comparable between the two groups.

Perceived oral health was linked to some functional difficulties such as toothache or difficulties chewing or talking. The proportion of participants who had difficulty talking increased proportionally with their self-perceived bad oral health in both the NT group (*p* = 0.040), but not in the OT group (*p* = 0.827). The same relationship was found with difficulty chewing in the NT group (*p* < 0.001), but not in the OT group (*p* = 0.105). Adolescents who felt embarrassed with the appearance of their teeth were those reporting worse self-perceived oral health, in both the NT group (*p* < 0.001) and in the OT group (*p* = 0.048). Adolescents having had toothache were those reporting bad self-perceived oral health in the NT group (*p* < 0.001), but not in the OT group (*p* = 0.145). The relationship between bleeding gums and self-perceived oral health was significant in the NT group (*p* < 0.001), but not in the OT group (*p* = 0.235).

## 4. Discussion

Studies on knowledge, attitudes and practices have been conducted at an international level in the field of public health for many years. This study, to our knowledge, represents the first study in France of this type which compares two groups of adolescents with or without orthodontic treatment.

We know today that adolescence is a critical period of transition where risky behaviours appear and that adolescent optimism is associated with positive health outcomes [32].

Orthodontists give oral health hygiene and dietary instructions before starting a treatment. We hypothesised that adolescents undergoing orthodontic treatment have better oral health knowledge, attitudes, and practices.

In this study, the knowledge of the adolescents was insufficient in both groups; except for the question about the role of fluoride in the prevention of caries, as adolescents undergoing treatment had a significantly higher percentage of correct answers (*p* < 0.001). 

Overall, 96.5% of adolescents without orthodontic treatment and 95.1% of adolescents with treatment knew that it is necessary to brush their teeth at least twice a day. In a study by Wahengbam et al. in 2016, adolescents’ knowledge about brushing twice a day was 86.4% [33].

Most participants reported having correct knowledge about the impact of snacking (78.5%) and sticky foods (94.7%) on the risk of caries. There was no significant difference between the two groups. Studies had similar results; 91.9% of adolescents knew about the role of diet in tooth decay in India [34,35,36,37], and this percentage was 86.3% in a recent study among adolescents from Portugal, Romania and Sweden [35]. 

Adolescents’ knowledge of fluoride was incorrect in both of the groups. However, adolescents with treatment had better results, mainly regarding the role of fluoride in the prevention of caries with 65.1% of correct answers, as opposed to adolescents without treatment, who had 46% correct answers (*p <* 0.001). Indeed, some studies had similar results in adolescents [33,35,36,37,38,39]. Adolescents undergoing orthodontic treatment had a better understanding of the role of fluoride.

Most adolescents in both groups (69%) claimed to brush their teeth twice a day. Adolescents undergoing orthodontic treatment brushed more. In the literature, the frequency of brushing in adolescents showed different results, from 40% [33] to 77.4% [35].

Adolescents undergoing orthodontic treatment use interdental brushes significantly more than other adolescents (35.1% vs. 4.4%) (*p* < 0.001). In addition, 20% of orthodontic patients in the study of R. Aljohani et al. used an interdental brush at least once daily [40]. Previous studies reported an infrequent use of interdental brushes (7–23%) among orthodontic patients [40,41,42]. A recent study carried out among adolescents aged 15 to 17 years old highlighted an average interdental inflammation of 96.5%. We now know that interdental spaces harbour pathogenic bacteria and that daily brushing reduces bleeding by 47% after 1 week of use and that symbiosis can be restored.

Our results show that very few adolescents did not consume fizzy drinks (3.5%). Many adolescents consumed fizzy drinks (48.7% and 39.8%). However, adolescents undergoing orthodontic treatment consumed significatively less (*p* = 0.040). In R. Graça et al.’s study, this concerned 35.5% of adolescents [35] and 56% in the study of R. Aljohani et al. [40]. 25.8% and 15.1% of adolescents—with and without orthodontic treatment, respectively—consumed fast-food (*p* = 0.03). For sweets, there were no significant differences between the two groups (33.6% and 24.7%), even though adolescents without orthodontic treatment tended to consume sweets more often. Results of between 40% and 60% can be found in the literature [35,36,40].

With regard to the use of drugs and alcohol, the results are similar in both groups. On the contrary, cigarette consumption is significantly higher among adolescents who do not have orthodontic treatment (*p* = 0.002). Additionally, 11.1% of adolescents without treatment and 4.2% of adolescents with treatment smoke on a regular basis. Smoking daily or occasionally was reported in 11.7% of adolescents in the study of T.Graça et al. [35].

This study has some limitations. Firstly, the adolescents were exclusively recruited in teaching dental hospitals (public health and orthodontic departments) and thus could not be fully representative of all adolescents. Particularly, those attending the dental public health department, and more generally dental care facilities, were more likely to have dental problems than the common population of adolescents. However, they were similar to the population attending the orthodontic department, regarding different factors such as socioeconomic or educational levels. This makes the comparison relevant in the context of different dental hospitals, drawn from different regions of the French territory.

Secondly, the structured questionnaire comprised the response “I don’t know” for most of the items. This option was supposed to help the respondent in sensitive choices, and thus avoid information bias as much as possible. The proportion of adolescents having chosen “I don’t know” may be very informative (lack of knowledge in prominent topics), nevertheless in some cases, we had to exclude them from the statistical tests, and sometimes very few respondents chose this option.

## 5. Conclusions

Adolescents lack oral health knowledge, and their attitudes and practices are not optimal. We know adolescence is an important period for the development of oral pathologies, and that this period is associated with hormonal changes, behavioural shifts, and changing eating habits. Adolescents start becoming more autonomous by making their own decisions. Orthodontists should be more aware of the need to teach their patients how to maintain good oral hygiene and have good dietary habits during orthodontic treatment to prevent cavities and periodontal disease. Oral health education is essential for all adolescents and the use of interdental brushes should be implemented.

## Figures and Tables

**Figure 1 ijerph-19-02316-f001:**
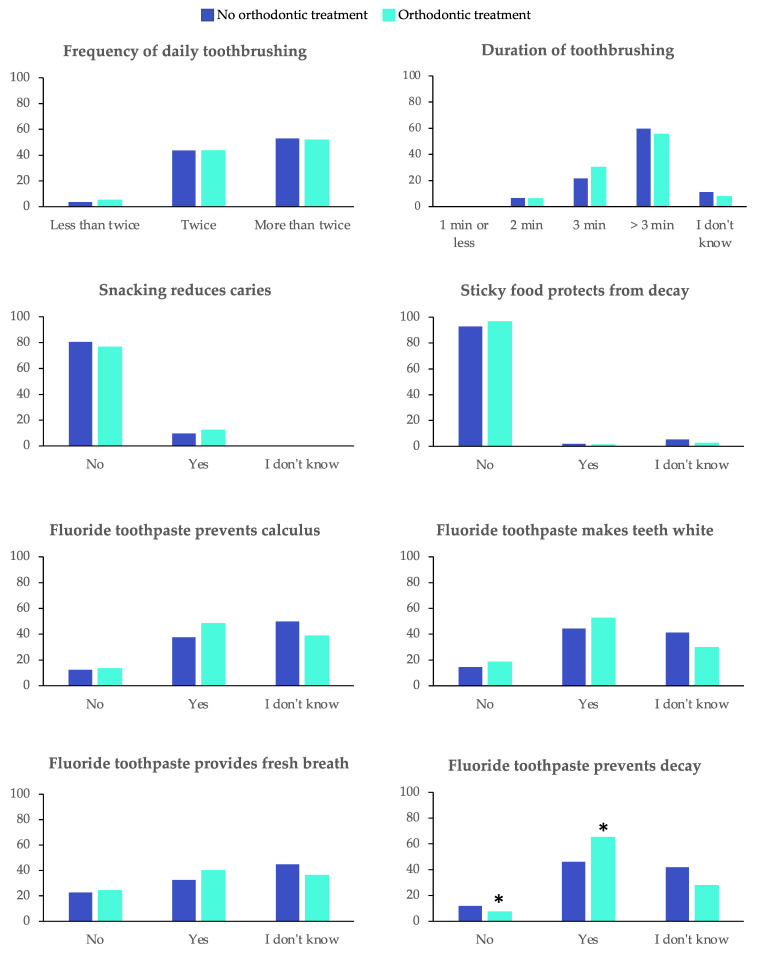
Knowledge of adolescents regarding prevention of oral diseases according to orthodontic treatment. * Statistically significance (*p* < 0.05). The p-values were calculated without considering the answer “I don’t know”.

**Figure 2 ijerph-19-02316-f002:**
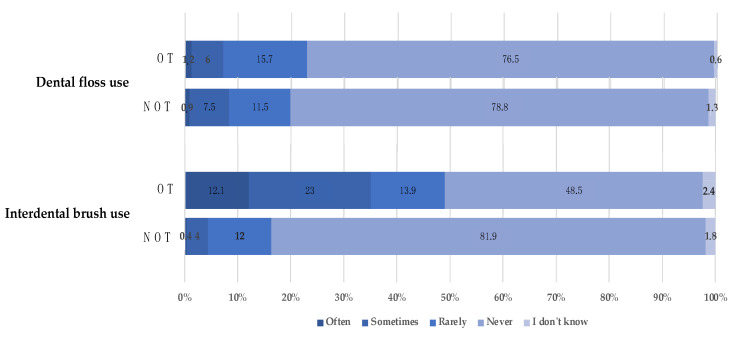
Oral hygiene habits among adolescents aged between 15 and 17 years old according to orthodontic treatment. The results are expressed as percentage. The OT group was composed of 266 adolescents and the NOT group was composed of 166 adolescents. OT: orthodontic treatment, NOT: no orthodontic treatment.

**Figure 3 ijerph-19-02316-f003:**
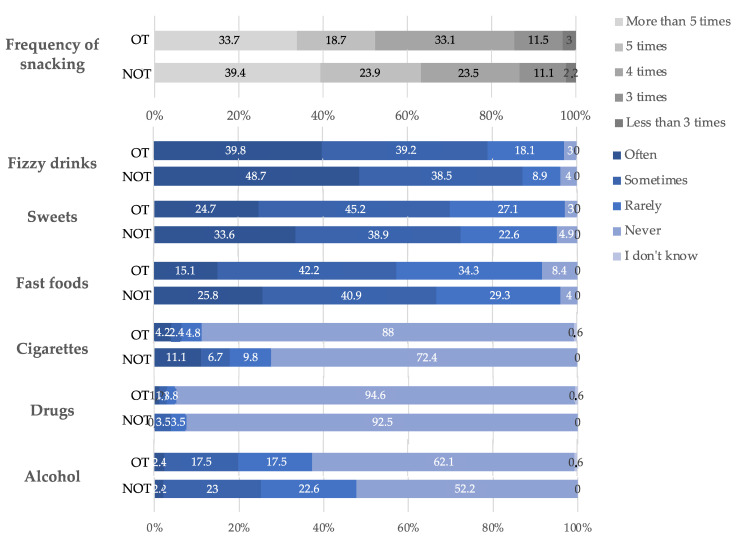
Nutrition and harmful habits among adolescents aged between 15 and 17 years old according to orthodontic treatment. The results are expressed as percentage. The OT group was composed of 266 adolescents and the NOT group was composed of 166 adolescents. OT: orthodontic treatment, NOT: no orthodontic treatment.

**Figure 4 ijerph-19-02316-f004:**
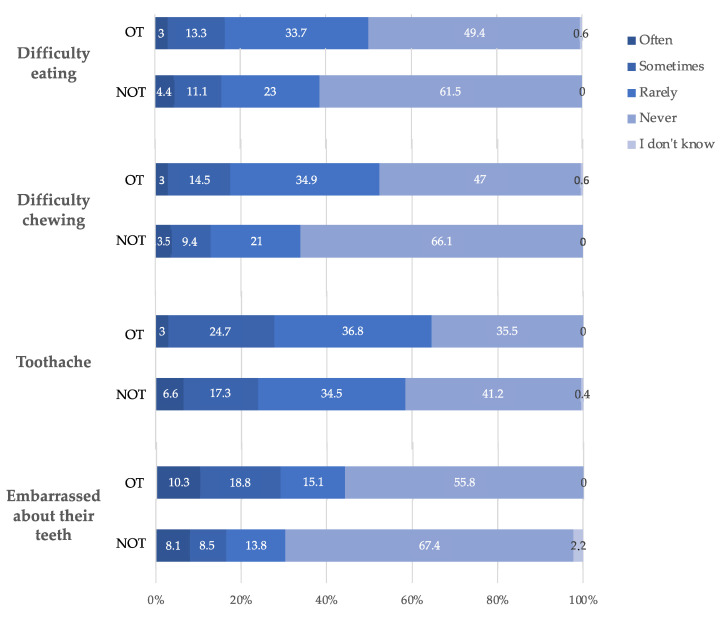
Self-perception of oral and general health according to orthodontic treatment. The results are expressed as percentage. The OT group was composed of 266 adolescents and the NOT group was composed of 166 adolescents. OT: orthodontic treatment, NOT: no orthodontic treatment.

**Table 1 ijerph-19-02316-t001:** Sociodemographic characteristics of the adolescents according to orthodontic treatment. n: number of adolescent, SD: standard deviation.

Variable		No Orthodontic Treatment (*n* = 266)	Orthodontic Treatment (*n* = 166)	*p*-Value (Chi-square)
Age	Years (Mean ± SD)	16.48 ± 0.53	16.38 ± 0.49	
Gender	Males (%)	56.6	53.6	0.552
	Females (%)	43.4	46.4	
Father’s occupation	Farm worker/Shopkeeper (%)	11.1	10.3	0.129
	Executive/Company manager (%)	24.8	30.7	
	Administration (%)	17.7	9.6	
	Employee/Manual worker (%)	29.6	37.4	
	Unemployed/Disabled/Deceased (%)	7.5	7.2	
	Unknown (%)	9.3	4.8	
Mother’s occupation	Farm worker/Shopkeeper (%)	5.7	3.6	0.060
	Executive/Company manager (%)	15.9	22.3	
	Administration (%)	29.7	23.5	
	Employee/Manual worker (%)	21.7	32.5	
	Unemployed/Disabled/Deceased (%)	19.5	15.7	
	Unknown (%)	7.5	2.4	
Any medical aid	No (%)	29.7	18.7	0.013
	Yes (%)	70.3	81.3	

## Data Availability

The data presented in this study are available on request from the corresponding author.

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
