# Peer review of "Oral Hygiene Practices and Knowledge among Adolescents Aged between 15 and 17 Years Old during Fixed Orthodontic Treatment: Multicentre Study Conducted in France"

_ijerph, 2022, doi:10.3390/ijerph19042316_

Round 1
Reviewer 1 Report
Oral hygiene practices and knowledge among adolescents aged between 15 and 17 years old during fixed orthodontic treatment: multicentre study conducted in France.
The purpose of this cross sectional survey was to assess oral health knowledge, attitudes and practices amongst a sample of 15-17-year-old French adolescents. The convenience sample was recruited from the public health and orthodontic departments of four teaching dental hospitals and allocated to one of two groups depending upon whether they were receiving orthodontic care or not. The study compared data from the two groups. Data demonstrated that while those receiving orthodontic care (and therefore regularly seeing an orthodontist) used an interdental brush more frequently than those who were not, drank fizzy drinks and ate fast foods less frequently, and were less likely to smoke cigarettes, over all oral health knowledge and practices among the sample were poor.
The inclusion and exclusion criteria were described and the sample size calculated. The sample of 392 adolescents exceeded the minimum sample of 380 although the groups were not equal, and the number of subjects in the orthodontic treatment group did not achieve the required minimum. The validated ESCARCEL survey was used to collect data.
Face-to-face interviews were conducted by members of the research team however line 113 identifies the researchers as (C.I., X.X., Y.Y., Z.Z., …) and needs to be corrected.
The statistical tests used in the study are described and are appropriate for the design and reflects items of questionnaire Appendix 1. The included tables are clear although in Table 4, results identify 0.6% respondents selected I don’t know to the items requesting information about use of cigarettes, drugs or alcohol, difficulty eating and chewing. These results suggest that respondents do not know if they smoke, take drugs or alcohol or have difficulty eating and chewing. If the result is correct, a short explanation is suggested to assist readers to interpret the data. The data supports the author’s conclusions and recommendations.
My recommendation is publish with minor revisions as suggested below:
1) Line 113: insert correct author’s initials (I., X.X., Y.Y., Z.Z., …)
2) Table 4: 0.6% respondents selected I don’t know to the items requesting information about use of cigarettes, drugs or alcohol, difficulty eating and chewing. If this is correct this results needs to be explained for readers.
Author Response
Dear reviewer 1,
Thank you for careful and thorough reading of this manuscript and for the thoughtful comments and constructive suggestions, which help to improve the quality of this manuscript. Our response follows (the reviewer’s comments are in black and our responses are in blue).
My recommendation is publish with minor revisions as suggested below:
1) Line 113: insert correct author’s initials (I., X.X., Y.Y., Z.Z., …)
We removed author’s initials. Because several dentists participated in the study.
2) Table 4: 0.6% respondents selected I don’t know to the items requesting information about use of cigarettes, drugs or alcohol, difficulty eating and chewing. If this is correct this results needs to be explained for readers.
We added line 111: “The response "I don't know" was selected in cases where the adolescent did not know the answer or did not wish to answer the question.”
Reviewer 2 Report
We can agree that oral health knowledge, attitude and practice during adolescent period of life may decide in future about risky behaviors and healthy choices in adult period of life activity. Therefore, exploration from France about orthodontic patients' self-reported oral health knowledge, attitude and practice appears important from public health point of view.
The authors may find the following comments.
1/ page 2, line 70 Aim of the study: replace word “describe” into “ evaluate “ or “ assess”
2/ table 1: what value do you show for variables gender, fathers’mothers’ occupations, and the rest? This is % or n=?
3/ p value in all places should have three numbers after the point, e.g. p=0.061 (not 0.06)
4/ my only criticism concerns into better presentation of data in graphical bars and figures, please make transition of tables 2,3,4.
5/ there is a lack of limitations of your study in Discussion Sector, could you find some?
Author Response
Dear reviewer 2,
Thank you for careful and thorough reading of this manuscript and for the thoughtful comments and constructive suggestions, which help to improve the quality of this manuscript. Our response follows (the reviewer’s comments are in black and our responses are in blue).
The authors may find the following comments.
1/ page 2, line 70 Aim of the study: replace word “describe” into “ evaluate “ or “ assess”
We made the modification
2/ table 1: what value do you show for variables gender, fathers’mothers’ occupations, and the rest? This is % or n=?
We modified the table to more clear because we indicated the percentage and only the total effectif of each group.
3/ p value in all places should have three numbers after the point, e.g. p=0.061 (not 0.06)
We made the modification
4/ my only criticism concerns into better presentation of data in graphical bars and figures, please make transition of tables 2,3,4.
We replaced Tables 2,3 and 4 by figures 1,2,3 and 4. The Tables 2,3 and 4 was added in supplementary file.
5/ there is a lack of limitations of your study in Discussion Sector, could you find some?
We added in discussion: This study has some limitations. Firstly, the adolescents were exclusively recruited in teaching dental hospitals (public health and orthodontic departments) and thus could not be fully representative of all adolescents. Particularly, those attending the dental public health department, and more generally dental care facilities, were more likely to have dental problems than the common population of adolescents. However, they were similar to the population attending the orthodontic department, regarding different factors such as socioeconomic or educational levels. This makes the comparison relevant in the context of different dental hospitals, drawn from different regions of the French territory.
Secondly, the structured questionnaire comprised the response “I don’t know” for most of the items. This option was supposed to help the respondent in sensitive choices, and thus avoid information bias as much as possible. The proportion of adolescents having chosen “I don’t know” may be very informative (lack of knowledge in prominent topics), nevertheless in some cases, we had to exclude them from the statistical tests, and some-times very few respondents chose this option.